# Total Tumor Diameter and Unilateral Multifocality as Independent Predictor Factors for Metastatic Papillary Thyroid Microcarcinoma

**DOI:** 10.3390/jcm10163707

**Published:** 2021-08-20

**Authors:** Liviu Hîțu, Paul-Andrei Ștefan, Doina Piciu

**Affiliations:** 1Doctoral School, Iuliu Hațieganu University of Medicine and Pharmacy, 400012 Cluj-Napoca, Romania; Liviu.Hitu@umfcluj.ro (L.H.); doina.piciu@gmail.com (D.P.); 2Anatomy and Embryology, Morphological Sciences Department, “Iuliu Hațieganu” University of Medicine and Pharmacy, Victor Babes Street 8, 400012 Cluj-Napoca, Romania; 3Radiology and Imaging Department, County Emergency Hospital, Clinicilor Street 5, 400006 Cluj-Napoca, Romania; 4Department of Endocrine Tumors and Nuclear Medicine, Institute of Oncology “Prof. Dr. Ion Chiricuță”, 400015 Cluj-Napoca, Romania

**Keywords:** papillary thyroid microcarcinoma, PTMC, total tumor diameter, TTD, unilateral multifocality, metastatic disease, independent predictors

## Abstract

**Simple Summary:**

Papillary thyroid microcarcinoma is currently the most frequent endocrine cancer at this time. Usually, this form of cancer is indolent, but there are situations in which it metastasizes. The current classification guidelines are rather simplistic and do not comprehend the whole disease spectrum. Studies that have addressed this issue have evaluated various stages of papillary thyroid carcinoma, considering the scarcity of studies based on European demographic data. We aim to further investigate whether total tumor diameter and multifocality are directly correlated with metastatic forms of papillary thyroid microcarcinoma. The results of this study could validate the confidence with which current guidelines are used or could open new avenues in using the total tumor diameter instead of the size of the largest tumor.

**Abstract:**

The purpose of this study was to assess whether total tumor diameter (TTD) and multifocality are predictors for metastatic disease in papillary thyroid microcarcinomas (PTMC). Eighty-two patients with histologically proven PTMC were retrospectively included. Patients were divided according to the presence of metastatic disease in the metastatic (*n* = 41) and non-metastatic (*n* = 41) demographic-matched group. The morphological features of PTMCs (primary tumor diameter, multifocality, TTD, number of foci, and tumor site) were compared between groups using univariate, multivariate, and receiver operating characteristic analyses. TTD (*p* = 0.026), TTD > 10 mm (*p* = 0.036), and Unilateral Multifocality (UM) (*p* = 0.019) statistically differed between the groups. The combination of the two independent predictors (TTD and UM) was able to assess metastatic risk with 60.98% sensitivity and 75.61% specificity. TTD and UM can be used to predict metastatic disease in PTMC, which may help to better adapt the RAI therapy decision. We believe that TTD and multifocality are tumor features that should be considered in future guidelines.

## 1. Introduction

Papillary thyroid microcarcinoma (PTMC) is defined as a malignant epithelial tumor with evidence of follicular differentiation and a series of specific nuclear features [1], with the maximum size of the tumor ≤ 1 cm [2]. The incidence of PTMC is increasing due to improved diagnostic methods such as ultrasound (US) with targeted fine-needle aspiration biopsy (FNAB) [2] and is estimated to account for more than 50% of new cases of thyroid cancer [3]. 

Although PTMC is considered to be the most indolent form of thyroid cancer, lymph node metastases (LNM) and local recurrence are frequently encountered [4]. The incidence rate of central LNM (CLNM) in PTMC is approximately 23–64.1%, and the incidence rate of lateral LNM (LLNM) in PTMC is approximately 3.7–44.5% [5,6,7]. 

Regardless of how comprehensive the content of the guidelines is, certain therapeutic settings are still limited. The Updated AJCC/TNM (American Joint Committee on Cancer/Union for International Cancer) Staging System for Differentiated and Anaplastic Thyroid Cancer (8th edition) defines primary tumor’s category only by the size of the greatest dimension [8]. The 2015 American Thyroid Association (ATA) places all intrathyroidal PTMCs, whether unifocal or multifocal, in the low-risk category. Only multifocal PTMCs with extrathyroidal extension (ETE) are considered to be in the intermediate-risk group [2]. Other international guidelines regarding thyroid cancer management do not have recommendations regarding PTMC treatment: National Comprehensive Cancer Network (NCCN) 2018 [9], European Thyroid Association (ETA) 2019 [1], and European Society for Medical Oncology (ESMO) 2019 [10]. 

Several studies show that multifocality/total tumor diameter (TTD) can better assess the aggressiveness of the tumor in PTMC [4,11,12,13,14,15,16]. Another study claims that calculating TTD in multifocal PTMC to evaluate adverse biological behavior is insufficient and limited [17]. Most research studies addressing the present topic of interest lack demographic data from Europe. Another essential point to note is that TTD has previously been assessed as a risk factor by comparing tumor size between different T stages of PTC (papillary thyroid carcinoma).

Our study aimed to find whether multifocality and TTD can function as predictors for metastatic disease in PTMC.

## 2. Materials and Methods

### 2.1. Conceptualization

TTDs’ impact on the development of metastatic disease in PTMC in the eastern European population was the focus of our research. This was possible by comparing a target group of metastatic PTMCs with a control group of PTMCs that did not have metastatic disease. To exclude as many aspects as possible that could bias the compared results, the non-metastatic group was chosen to have epidemiological characteristics as near as possible to the target group.

### 2.2. Study Design and Population

This retrospective study was approved by the Ethical Committee of “Iuliu Hatieganu” University of Medicine and Pharmacy, Cluj-Napoca (number 4458) and of “Prof. Dr. Ion Chiricută” Institute of Oncology, Cluj-Napoca (number 175/5). The data collection was done retrospectively including all patients treated in our regional oncological center between January 2008 and March 2021 that met the following criteria: initial surgical management of the thyroid, total thyroidectomy associated with central neck dissection, full clinicopathological information available, and final pathological diagnosis of PTMC. Patients with coexisting malignancies, previous history of radiotherapy to the head and neck region, and incomplete data were excluded. All patients have signed the institutional informed consent on participation in scientific studies.

### 2.3. Data Collection

We conducted a search in our institution database using the keywords: papillary+ thyroid + microcarcinoma + metastasis. After applying the selection criteria, 41 patients with metastatic PTMC were identified. Using the metastatic group’s demographic data, we found another 41 patients with PTMC who underwent lymph node dissection but did not have metastatic disease and had demographic parameters that were as close to the metastatic group as possible.

Data collection regarding demographic characteristics, the diagnosis, and the therapeutic protocol was retrieved from the patient medical file. The histopathological information was extracted from the original pathology report. Papillary tumors measuring 1 cm or less in diameter were defined as PTMCs. TNM grading was performed according to the 8th edition of the TNM classification introduced by the American Joint Committee on Cancer [8]. The pathologic features examined were central and lateral nodal metastasis, microscopic and gross extrathyroidal extension (ETE), lymphovascular invasion (LVI), and distant metastasis. A tumor was defined as multifocal if at least 2 foci were found. Multifocality/Unifocality was divided into four separate entities: Unilateral Multifocality (2 or more foci in the same lobe); Bilateral Multifocality (more than 1 focus in both lobes); Bilateral Unifocality (1 focus in each lobe) and Unilateral Unifocality (unique focus). For multifocal lesions, the sum of the maximal diameter of each tumor foci was used to calculate TTD. Patients were divided into two age groups according to age at the time of diagnosis, <55 years and older.

### 2.4. Statistical Analysis

The metastatic group characteristics were compared with the non-metastatic group. For categorical variables, the Chi-square test and Fisher test were used. For continuous variables, the distribution was tested through the Kolmogorov–Smirnov Test of Normality. For normal distributed continuous variables- independent samples *t*-test was used, and for non-normal distributed continuous variables—Mann–Whitney U-test. Multivariate regression analyses were performed to identify independent risk factors for metastases. A *p*-value < 0.05 was considered statistically significant.

We investigated which of the parameters that showed statistically significant results at the univariate analysis are also independent predictors of metastases. In this regard, a multivariate regression analysis (using the “enter” input model) was conducted, with the computation of the coefficient of determination (R-squared) and the variance inflation factor (VIF). Since a high VIF value is an indicator of multicollinearity, features that recorded a VIF of ≥10^4^ were excluded from further analysis. The predicted values were saved and subsequently used in a receiver operating characteristics (ROC) analysis to assess the diagnostic power of the entire prediction model. The ROC analysis was also used to determine the diagnostic power of features independently associated with metastases, along with the calculation of the area under the curve (AUC), sensitivity, and specificity, with 95% confidence intervals (CIs). Optimal cut-off values were chosen using a common optimization step that maximized the Youden index for predicting patients with metastatic disease. Sensitivity (Se) and specificity (Sp) were computed from the same data, without further adjustments. Statistical analysis was performed by an independent statistician, using The Statistical Package for Social Sciences software (SPSS, version 22.0, Chicago, IL, USA) and MedCalc version 14.8.1 (MedCalc Software, Mariakerke, Belgium).

## 3. Results

### 3.1. Baseline and Tumoral Characteristics

Baseline clinicopathological characteristics of the 82 patients who underwent thyroidectomy due to PTMC are presented in Table 1. Mean age of the study participants was 45.5 years with a standard deviation (SD) of 14.0; among these patients 60 (73.2%) were women. Patients younger than 55 years old had a mean age of 37.8 years with a SD of 9.6, while those older than 55 years old had a mean age of 62.0 years with a SD of 4.9. Regarding pN staging, 42 patients (51.2%) were staged N0, 29 patients (35.4%) were N1a staged and 11 patients (13.4%) were staged N1b. Of all patients included in the study, there was only one case of distant metastasis (1.2%)—in the left gluteus muscle. The number of patients in each TNM stage was as follows: 69 (84.2%) in stage I, 12 (14.6%) in stage II, and 1 (1.2%) in stage IVb.

Analyzing the distribution of PTMC subtypes, 52 patients (63.4%) had conventional PTMC, the second most common subtype being the follicular variant-23 patients (28.1%). Four patients (4.9%) had oncocytic variant, one (1.2%) for diffuse sclerosing, one (1.2%) for solid variant and one (1.2%) for columnar cell variant.

The lymphatic invasion was present for 13 patients (15.9%), vascular invasion for 8 patients (9.8%), and perineural invasion was observed for only 5 patients (6.1%). A microscopic capsular invasion was found in 24 patients (29.3%). Table 2 shows the focal and dimensional features. The median value for primary tumor diameter (PTD) was 5.0 mm with an interquartile range (IQR) of 5.3 mm. The mean value for PTD less than 5 mm was 4.0 with an IQR of 1.5, whereas the mean value for PTD 6–10 mm was 9.0 with a 2.5 IQR.

Multifocality was found in 45 patients (54.9%), with 14 (17.0%) having numerous foci in a single lobe (unilateral multifocality) and 18 patients (22.0%) having multifocality in both lobes (bilateral unifocality). Unifocality was identified in 37 patients (45.1%), with 13 (15.9%) patients having it in both thyroid lobes (bilateral unifocality) and 14 (17.0%) patients having a single focus (unilateral unifocality).

The total tumor diameter (TTD) median was 7.75 mm with a 6.4 IQR, for the group with TTD ≤ 10, the median was 5.0 mm with 6.0 IQR, and for the >10 TTD group the median was 14.0 mm with a 9.0 IQR. There were 37 patients (45.1%) with a single tumoral focus, 18 patients (22.0%) with two tumoral foci, 13 patients (15.9%) with three tumoral foci, and 14 patients (17.0%) with four or more tumoral foci. The largest fraction of patients had tumor localization in both the right and left lobes-29 patients (35.4%), followed by the right thyroid lobe site for 27 patients (32.9%), left lobe for 18 patients (35.4%), and isthmus ± other location for 8 patients (9.7%). 

### 3.2. Comparison between Metastasis and No Metastasis Groups

Gender and age are almost identical characteristics and therefore will not be described comparatively. Naturally, the non-metastatic group is all N0 staged. However, there was one patient in the metastatic group who is classed as N0 (a patient staged M1- with a distant solitary muscle metastasis, in the left gluteus muscle). The patient was operated by total thyroidectomy at the end of 2009. For 8 years, the patient was in complete remission and disease-free. In 2018, thyroglobulin level started to rise, the patient received a dose of radioactive iodine, with a negative post-therapy I-131 whole-body scan. For further evaluation, a F-18 fluorodeoxyglucose (FDG) positron emission tomography/computer tomography (PET/CT) scan was performed, which showed a 39/35/41 mm tumor in the left gluteal muscle with focal pathological uptake SUV lbm max = 16.77, highly suggestive for a metastatic lesion. After surgery and histology exam, the results confirmed papillary thyroid carcinoma metastasis. The patient received another I-131 dose of 5.5 GBq, with negative WBS, and was submitted to external beam therapy; at the moment of writing this paper, the patient was alive and clinically negative [18].

The rest of the patients in the metastatic group were in stages N1a-29 patients (70.7%) and N1b-11 patients (26.8%). All patients in the non-metastatic group were classified as stage I according to AJCC. In the metastatic group, 28 patients (68.3%) were defined as stage I, 12 (29.3%) as stage II, and one (2.4%) as stage IVb (same patient with pN0M1 staging mentioned above). In terms of PTMC subtype, lymphatic, vascular, perineural, and microscopic capsular invasion, no statistically significant differences were identified between the two groups (Table 1).

There was a statistically significant difference between the two groups in terms of multifocality and TTD. The proportion of patients in the metastatic group with unilateral multifocality was significantly higher than in the non-metastatic group (26.8% vs. 7.3%, *p* = 0.019). Based on TTD, the metastatic group had a considerably higher dimension compared to the non-metastatic group (median ± IQR: 9.0 ± 7.2 vs. 5.0 ± 7.0 mm., *p* = 0.026). Furthermore, there was a significant difference between the metastatic group with TTD > 10 mm. compared to the non-metastatic group (median ± IQR: 17.0 ± 12.3 vs. 12.0 ± 3.9 mm., *p* = 0.036).

### 3.3. Predictors for Metastatic Disease

A multivariate analysis was used to identify which of the statistically significant characteristics may be used as an independent predictor of metastatic disease. The multivariate analysis showed a significant level of *p* < 0.0026, an R2 coefficient of determination of 0.1663, an adjusted R^2^ of 0.1342, and a multiple correlation coefficient of 0.4078 (Table 3). TTD and UM were found to be independent predictors of metastatic disease in PTMC, whereas TTD > 10 was not statistically significant. For additional statistical research, a prediction model was created. TTD, UM, and the prediction model were subjected to a ROC analysis (Table 4, Figure 1). 

The cut-off value for TTD of >4.4 mm was found to be an independent predictor of metastatic disease in PTMC (*p* = 0.0197, Se = 78.05%, Sp = 46.34%). The presence of UM was also shown to be an independent predictor (*p* = 0.0163, Se = 26.83%, Sp = 92.68%). The statistical characteristics of TTD and UM were translated by a prediction model with the following statistical values (*p* < 0.0001, Se = 60.98%, Sp = 75.61%). 

## 4. Discussion

TTD/multifocality in PTMC have been the subject of several recent papers [11,12,13,14,15,16,17,19,20,21]. Most of the research was conducted on a large cohort of patients and provides extremely useful data; nevertheless, European demographic data are scarce, with the majority of studies focusing on Asian populations [11,12,13,14,15,16,17,19,20] and one in North America [21]. Furthermore, some of these studies compare risk factors for PTMC and PTC groups, although tumor sizes vary widely and comparative terms can be frequently misunderstood.

In our research, multifocality was found in 54.9% of PTMCs and 48.8% had LNM. There was no statistical difference between the metastatic and non-metastatic groups in terms of multifocality (*p* = 0.270). This result contradicts the findings of most research, which demonstrate a link between multifocality and metastatic disease [4,5,6,11,12,14]. This contradictory result may be a consequence of the different cohorts (in our study the number of patients in the two groups is equal vs. the other studies that have a much higher number of patients in the non-metastatic group).

Alternatively, a statistically significant difference in unilateral multifocality was observed in our study (26.8% vs. 7.3%, *p* = 0.019). Unilateral multifocality was also shown to be an independent predictor of metastatic disease in our study. Similar to our results, according to Cai et al. [20], patients with unilateral multifocality were more likely than those with bilateral multifocality to develop neck metastases. In contrast, the results published by Yan et al. [19] show that bilateral multifocality, rather than unilateral multifocality, should be considered as an aggressive marker at presentation, and neither is an independent prognostic factor for clinical outcome in PTMC. 

When we investigated the TTD, the findings of our research indicated that the metastatic group had a considerably higher dimension compared to the non-metastatic group (median ± IQR: 9.0 ± 7.2 vs. 5.0 ± 7.0 mm, *p* = 0.026). In addition, there was a significant difference between the metastatic group with TTD > 10 mm. compared to the non-metastatic group (median ± IQR: 17.0 ± 12.3 vs. 12.0 ± 3.9 mm., *p* = 0.036). Similar findings were revealed in research published by Feng et al. [11]. The results of his study showed that multifocal PTMC with TTD > 10 mm was more aggressive than unifocal PTMC or multifocal PTMC with TTD ≤ 10 mm. Likewise, the results of Zhao et al. [12] showed that LNM frequency was significantly higher in multifocal PTMCs with TTD > 10 mm than unifocal tumors with a diameter ≤ 10 mm (60.4 vs. 30%, *p* < 0.001). 

According to Liu et al. [15], the risks of LNM, extrathyroidal extension (ETE), infiltration, and the recurrence-free survival were significantly different between PTMCs with a unifocal diameter ≤ 10 mm and multifocal TTD > 10 mm and between multifocal PTMCs with a TTD of ≤1 mm and >10 mm. TTD might be used as a criterion to identify individuals at increased risk of persistence, according to Tam et al. [16], and T1a multifocal PTMCs with TTDs of 1 to 2 cm might be classed as T1b. However, there are also published data that demonstrate that calculating the TTD to assess adverse biological behavior in multifocal PTMC is insufficient and limited [17].

The novel findings of our research showed TTD and UM as independent predictors of metastatic disease in PTMC. The cut-off value of TTD > 4.4 mm independently predicts metastatic disease with a Se of 78.05% and Sp of 46.34%. On the other hand, the presence of UM independently predicts metastatic disease in PTMC (Se = 26.83%, Sp = 92.68%). Integrating TTD and UM statistical characteristics, a prediction model for metastatic disease has been developed (Se = 60.98%, Sp = 75.61%). 

The latest consensus statements regarding the strategy for active surveillance of adult low-risk PTMC published by Sugitani et al. [22] on behalf of the Japan Association of Endocrine Surgery Task Force on management for papillary thyroid microcarcinoma consider that no data suggest that tumor multiplicity is associated with tumor enlargement and appearance of LNM; thus, patients with PTMC and multiple lesions can be candidates for active surveillance. Our data and results suggest a special precaution related to multiplicity, the UM being in our cohort an independent factor that predicts metastatic disease.

ATA guidelines [2] do not indicate routinely the Radioactive Iodine Therapy in PTMC, except the association of aggressive histology or other specific individual conditions (ex., discordant thyroglobulin level after surgery, etc.) Considering the abovementioned results, the Radioiodine therapy decision might be better adjusted.

There are several drawbacks to this study. The small number of patients included in the research is one of the limitations. This is since prophylactic lymph node dissection (LND) is not performed routinely in our center, thus the number of patients with a histopathological result of PTMC that includes the status of lymph nodes being very limited. Furthermore, this is a retrospective research based on a single regional center’s experience. In light of this, randomized case-control clinical multicenter studies are required.

## 5. Conclusions

Regardless of how comprehensive the content of the guidelines is, certain therapeutic settings remain insufficiently evaluated. Our data strongly indicate that TTD and UM can be used to predict metastatic disease in PTMC, which may help to better adapt the RAI therapy decision. We believe that TTD and multifocality are tumor features that should be considered in future guidelines.

## Figures and Tables

**Figure 1 jcm-10-03707-f001:**
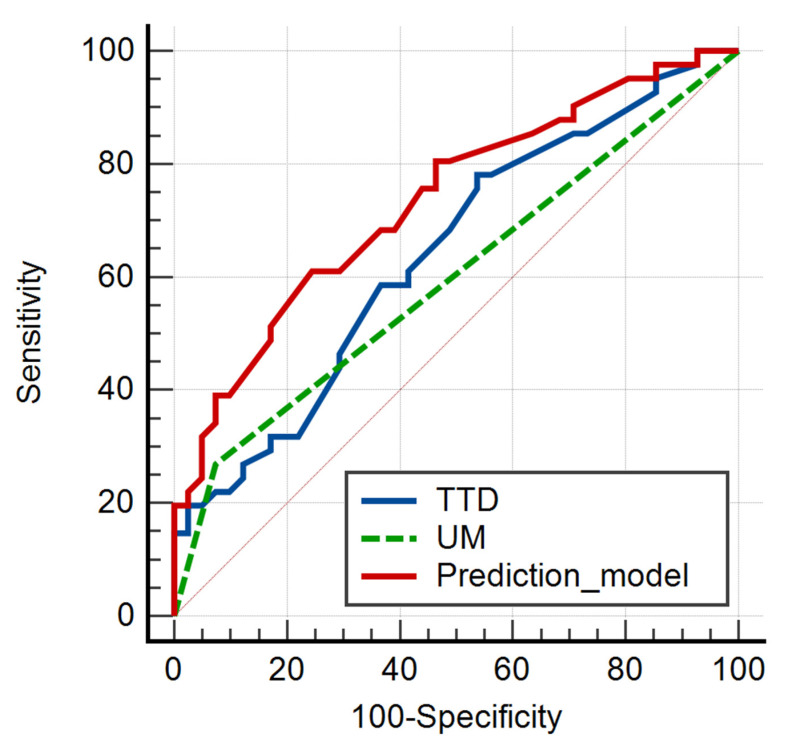
Receiver operating characteristic curve of the two parameters independently associated with the presence of PTMC metastatic disease and the prediction model. TTD-total tumor volume; UM-unilateral multifocality.

**Table 1 jcm-10-03707-t001:** Clinicopathological characteristics and the univariate analysis.

	No Metastatic Disease	Metastatic Disease	Total	*p*-Value
gender, *n (%)*				1
male	11 (26.8%)	11 (26.8%)	22 (26.8%)	
female	30 (73.2%)	30 (73.2%)	60 (73.2%)	
age at diagnosis (years)				
mean ± SD	46.5 ± 13.5	44.5 ± 14.7	45.5 ± 14.0	0.524
age group, (years)				
<55 (mean ± SD)	39.1 ± 8.9	36.5± 10.3	37.8 ±9.6	0.310
≥55 (mean ± SD)	62.3 ± 5.8	61.7 ± 3.9	62.0 ±4.9	0.754
pN stage, *n (%)*				
N0	41 (100%)	1 (2.4%)	42 (51.2%)	**0.0001**
N1a	0	29 (70.7%)	29 (35.4%)	
N1b	0	11 (26.8%)	11 (13.4%)	
M stage, *n* (%)				**0.0001**
M0	41 (100%)	40 (97.6%)	81 (98.8%)	
M1	0	1 (2.4%)	1 (1.2%)	
AJCC TNM staging, *n (%)*				
I	41 (100%)	28 (68.3%)	69 (84.2%)	**0.0001**
II	0	12 (29.3%)	12 (14.6%)	
IVb	0	1 (2.4%)	1 (1.2%)	
PTMC subtype, *n* (%)				
Conventional	25 (61.0%)	27 (65.9%)	52 (63.4%)	0.532
Follicular variant	12 (29.3%)	11 (26.9%)	23 (28.1%)	
Oncocytic	3 (7.3%)	1 (2.4%)	4 (4.9%)	
Diffuse sclerosing	0	1 (2.4%)	1 (1.2%)	
Solid/Trabecular	0	1 (2.4%)	1 (1.2%)	
Columnar cell	1 (2.4%)	0	1 (1.2%)	
lymphatic invasion, *n (%)*				0.364
presence	5 (12.2%)	8 (19.5%)	13 (15.9%)	
absence	36 (87.8%)	33 (80.5%)	69 (84.1%)	
vascular invasion, *n (%)*				1
presence	3 (7.3%)	5 (12.2%)	8 (9.8%)	
absence	38 (92.7%)	36 (87.8%)	74 (90.2%)	
perineural invasion, *n (%)*				0.712
presence	2 (4.9%)	3 (7.3%)	5 (6.1%)	
absence	39 (95.1%)	38 (92.7%)	77 (93.9%)	
microscopic capsular invasion, *n (%)*				1
presence	12 (29.3%)	12 (29.3%)	24 (29.3%)	
absence	29 (70.7%)	29 (70.7%)	58 (70.7%)	

*n*—data expressed as patients number (%); pN—pathologic lymph node stage; M—distant metastasis; AJCC TNM—American Joint Committee on Cancer Classification of Malignant Tumors; PTMC—papillary thyroid microcarcinoma. A statistically significant difference was defined as *p* < 0.05; bold values are statistically significant.

**Table 2 jcm-10-03707-t002:** Focal and dimensional characteristics and the univariate analysis results.

	No Metastatic Disease	Metastatic Disease	Total	*p*-Value
primary tumor diameter (mm)				
median ± IQR	4.0 ± 4.5	6.0 ± 5.0	5.0 ± 5.3	0.061
≤5 mm	3.2 ± 2.0	4.0 ± 1.0	4.0 ± 1.5	0.445
6–10 mm	9.0 ± 2.0	9.0 ± 2.6	9.0 ± 2.5	0.453
multifocality, *n* (%)				
presence	20 (48.8%)	25 (61.0%)	45 (54.9%)	0.270
absence	21 (51.2%)	16 (39.0%)	37 (45.1%)	
bilateral unifocality	8 (19.5%)	5 (12.2%)	13 (15.9%)	0.364
bilateral multifocality	9 (22.0%)	9 (22.0%)	18 (22.0%)	1
unilateral unifocality	21 (51.2%)	16 (39.0%)	37 (45.1%)	0.267
unilateral multifocality	3 (7.3%)	11 (26.8%)	14 (17.0%)	**0.019**
TTD (mm)				
median ± IQR	5.0 ±7.0	9.0 ± 7.2	7.75 ± 6.4	**0.026**
≤10	4.0 ± 5.0	6.0 ± 5.0	5.0 ± 6.0	0.059
>10	12.0 ± 3.9	17.0 ± 12.3	14.0 ± 9.0	**0.036**
number of foci, *n* (%)				
1	21 (51.2%)	16 (39.0%)	37 (45.1%)	0.267
2	9 (22.0%)	9 (22.0%)	18 (22.0%)	1
3	6 (14.6%)	7 (17.0%)	13 (15.9%)	0.762
≥4	5 (12.2%)	9 (22.0%)	14 (17.0%)	0.240
tumor site, *n* (%)				
RTL	13 (31.7%)	14 (34.1%)	27 (32.9%)	0.814
LTL	9 (22.0%)	9 (22.0%)	18 (22.0%)	1
RTL + LTL	17 (41.5%)	12 (29.3%)	29 (35.4%)	0.248
isthmus ± other location	2 (4.9%)	6 (14.6%)	8 (9.75%)	0.139

*n*—data expressed as patients number (%); IQR—interquartile range; TTD—total tumor diameter; RTL—right thyroid lobe; LTL—left thyroid lobe. A statistically significant difference was defined as *p* < 0.05; Bold values are statistically significant.

**Table 3 jcm-10-03707-t003:** Multivariate analysis results showing the characteristics independently associated with metastatic disease in PTMC. Bold values are statistically significant.

Least Squares Multiple Regression						
Sample size		82				
Coefficient of determination R^2^		0.1663				
R^2^-adjusted		0.1342				
Multiple correlation coefficient		0.4078				
Residual standard deviation		0.4681				
Regression Equation						
Independent variables	Coefficient	Std. Error	t	*p*	r_partial_	r_semipartial_
(Constant)	0.2297					
TTD	0.03242	0.01130	2.868	**0.0053**	0.3089	0.2965
UM	0.3312	0.1375	2.409	**0.0183**	0.2632	0.2491
TTD > 10	−0.2383	0.1707	−1.396	0.1666	−0.1561	0.1443
Analysis of Variance						
Source	DF	Sum of Squares			Mean Square	
Regression	3	3.4090			1.1363	
Residual	78	17.0910			0.2191	
F-ratio	5.1861					
Significance level	***p* = 0.0026**					

**Table 4 jcm-10-03707-t004:** The receiver operating characteristic analysis results of the parameters that are independently associated with the presence of PTMC metastatic disease and the prediction model consisting of these parameters. Between the brackets are the values corresponding to the 95% confidence interval.

Parameter	AUC	Significance Level	J	Cut-Off	Se (%)	Sp (%)
TTD	0.642	0.0197	0.2439	>4.4	78.05	46.34
	(0.529 to 0.745)				(62.4–89.4)	(30.7–62.6)
UM	0.598	0.0163	0.1951	>0	26.83	92.68
	(0.483 to 0.704)				(14.2–42.9)	(80.1–98.5)
Prediction model	0.734	<0.0001	0.3659	>0.4890	60.98	75.61
	(0.625 to 0.826)				(44.5–75.8)	(59.7–87.6)

TTD—total tumor diameter; UM—unilateral multifocality; AUC—area under curve; Se—sensitivity; Sp—specificity.

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
