# Peer review of "Total Tumor Diameter and Unilateral Multifocality as Independent Predictor Factors for Metastatic Papillary Thyroid Microcarcinoma"

_jcm, 2021, doi:10.3390/jcm10163707_

Round 1
Reviewer 1 Report
In this study the authors investigate whether total tumor diameter and multifocality may represent predictor factors for metastatic disease in papillary thyroid microcarcinomas . They compare clinical-pathological features of 82 microPTCs stratified in two equally distributed and demographic-matched groups according to the presence/absence of metastases.
Eventhoght monocentric and based on a small number of patients, the study could be of interest. However, some points require to be better addressed:
- the authors refer to the general term “metastatic disease”, but the study appears instead to be designed for the assessment of predictors related to lymph nodes (LN) metastatic disease. The presented caselist indeed include 40 patients with LN metastases and just one patient with solitary muscle metastasis (line 176). Which is the metastatic site for this latter? Whether it is a distant metastasis this should be clearly indicated in the text and also in Table1. This patient seems to represent an outlier in this caselist, thus additional evaluations should be included, performing the same analyses also excluding this patient.
-Histological variants represent a relevant issue in PTC as well as in microPTC. The samples’ histological variant should be reported, as well as the analysis of its distribution in the two groups and whether a specific association with TTD and/or UM exists.
Author Response
The answer was submitted in the attached word document.

Reviewer 2 Report
The authors have highlighted findings identified in a European demographic that led to the identification of key characteristics such as TTD and UM to predict metastatic disease which are significant to the field of Thyroid Cancer. The manuscript needs minor revisions and rigorous editing in terms of language and proofreading.
Although it has hard to conclude that 82 patients (data from a decade) from one Institute represents the entire European demographic, the current study is of significance more so due to data insufficiency in the region. It would be ideal if this study included patient data from other institutes in Europe. Nevertheless, their study and findings within the limitations are of relevance to the current field of study.
Author Response
The answer was submitted in the attached word document

Round 2
Reviewer 1 Report
The revised version of the manuscript has been improved.
The requested points about the description of distant metastasis patient and about histological variants have been addressed. It could be informative for distant metastasis patient (line 182-193) to specify also the corresponding histological variant.
The requested additional evaluations comparing no metastatic patients vs. LFN metastatic patients (i.e. excluding the single distant metastatic patient) however are still lacking.
This manuscript is a resubmission of an earlier submission. The following is a list of the peer review reports and author responses from that submission.
Round 1
Reviewer 1 Report
The authors have evaluated series number of clinico-pathological components in metastatic and non-metastatic disease of miPTC by grouping micPTCs as Unifocal (UU), multifocal (UM and BM).
The idea is very nice and timely. I have some questions/concerns about the conclusion made:
First of all, the number of the cases are very limited for such daring conclusion. Micro PTCs generally are very common ,in any population even with the applied criteria of the study. Case numbers needs to be greatly enlarged.
Second concern is about the description of grouping the multifocality.
UM- is 1 focus in each lobe
BM-is more than 1 focus in each lobe.
How about cases more than 1 focus in 1 lobe? and the cases that have 1 focus in 1 lobe while having more than 1 focus in other lobe?
If those cases were not included, was there any exact scientific reason or the case cohort simply did not have these possibilities ? Without those cases/possibilities are involved, these conclusions are too far to be made.
Third and the last, how come TTD is an independent predictive factor for metastatic miPTC and there is no any statistical significance with BM while showing the significance with UM? TTD's in UM was higher than TTD's in BM? Does authors think any possible biological or scientific reason behind this? Or just the limited numbers of the case cohort?
Reviewer 2 Report
The aim of the paper is to show if the total tumor diameter and multifocality of the papillary thyroid microcarcinoma(PTMC) are predictors for metastatic disease. The assessment of the metastases has been evaluated exclusively on the histolgical response of the removed lymph nodes performing the central neck dissection. No information on how the preoperative diagnosis of PTMC has been achieved is referred in the paper. This could be an important parameter to clarify because it is different the surgical approach and the prognosis for clinically discovered PTMC and incidental PTMC. Also the familiarity for papillary carcinoma could influence the results and should be indicated. Another point that is not indicated in the paper is the site of the PTMC (at least the greatest nodule) in the thyroid lobe. The localization of the tumor in middle part of the lobe or in the isthmus could influence the positivity of central neck lymph nodes in relationship to the direction of the lymphatic flow. Furthermore, the histogical description of PTMC is not completely detailed: are there present in the case series also noninvasive follicular thyroid neoplasm with papillary-like features(NIFTP)? Have been considered as metastases also the lymph nodes micrometastases? Finally, a comment on the value of the prediction model based on the two histological parameters considered by the Authors as significant should be provided. Are a sensitivity of 60% and a specificity of 75% considered satisfactory by the Authors?
Reviewer 3 Report
The study by Liviu HîÈ›u et al. requires some changes before it can be considered for publication:
Section keywords: Multi-docality – a misspelled word
Introduction section - Papillary thyroid microcarcinoma (PTMC) - more detailed tumor characteristics, such as subtypes of cancer, perhaps the presence of mutations (if possible), aggressivity and larger group size are necessary to make the results reliable
Table 1 – there’s a problem with the table’s legend
Section 3.1 A microscopic capsular invasion was found in 18 patients (22.0%). – this data doesn’t match the data from the table 1
Table 3, section 3.3, the conclusion section
- TTD and UM were found to be independent predictors of metastatic disease in PTMC, whereas TTD> 10 was not statistically significant.
- The cut-off value for TTD of > 4.4 mm was found to be an independent predictor of metastatic disease in PTMC
- We believe that TTD and multifocality are tumor features that should be considered in future guidelines
In my opinion authors should reconsider using TTD as a predictor of metastatic disease, because the results showing that TTD > 4.4 mm, but not larger than 10mm sound very untrustworthy. The results of AUC in the ROC analysis for TTD of 0.6 just barely support the theory of TTD being a predictor. Another factor is different results for TTD > 10mm in univariate and multivariate analyses.
Only the UM has been proven as a predictor of metastatic disease, but the comparison of 3 vs 11 patients supporting this finding still calls for a larger group size